# Robust control of Robotic Manipulator with Prescribed Spatio-temporal and Input constraints

*Abstract*— Precise tracking of a robotic arm motion subject to exogenous disturbances within a stipulated time is a challenging problem. Existing methods either rely on adaptive laws based on structured parameterizations or disturbance observers, which are sensitive to initial parameter estimates. To mitigate these limitations, we propose a robust adaptive control policy that achieves the prescribed time convergence within predefined bounds while adhering to state and input constraints in the presence of unknown external disturbances. In particular, we formulate filtered errors using time-based generators to prespecify settling time. Design a constraint law to enforce state constraints and impose input limits via a saturation function. Then, an adaptive barrier-function-based controller is synthesized from the filtered errors to achieve prescribed time stability. Moreover, the proposed control framework directly enforces state and input constraints in the control law, avoiding Lyapunov redesign required by barrier Lyapunov function methods and the optimization-based filtering used in control barrier function approaches. Prescribed-time stability of the closed-loop system is rigorously established via Lyapunov analysis. Finally, extensive experiments [1] are conducted on the Franka Robotic arm under various external disturbances to demonstrate the efficacy of the proposed scheme.

## I. Introduction

Control of robotic manipulators in the presence of external disturbances could lead to poorer tracking performance. In particular, payload variation due to a sudden change in payload, leading to impulsive behavior or dynamic interaction with the environment, is difficult to model as structured disturbances. Moreover, control policies may become under- or over-approximate due to inaccurate information about model parameters, or the model itself often leads to inaccurate tracking performance. Thus, synthesizing control policies for systems such as robotic manipulators in the presence of model uncertainties and unknown bounded disturbances to achieve accurate tracking helps in solving practical challenges in industrial automation, human–robot interaction, and contact-rich manipulation tasks.

To address this problem, adaptive control strategies estimate uncertain dynamics online and incorporate these estimates into the control policy to drive tracking errors to zero. Such methods are typically developed using either linear parametrization of the dynamics or lumped uncertainty formulations. In the former approach [1], [2], [3], [4], the dynamics are represented by a regressor matrix and parameter vector for online adaptation; however, deriving this parametrization becomes tedious for high-dimensional systems such as 7-DoF manipulators, where up to $10n$

[1]Video link: https://drive.google.com/file/d/1w_kO3HgBdsdw1wD-Z0hvKNy_2b8zpg2q/view?usp=drive_link

(n=7) inertial parameters may be required [5]. In the latter approach, unknown dynamics are grouped into a lumped uncertainty term and estimated using adaptive laws, function approximators, or disturbance observers [6], [7], [8], [9], [10], [11], [12], [13]. Nevertheless, both frameworks remain sensitive to the initialization of adaptive parameters and disturbance estimates, often affecting transient tracking performance during the early adaptation phase, particularly under significant model uncertainty [14], [15], [16], [17].

Furthermore, compliance with time-domain constraints is critical in applications such as factory-floor operations, where tasks must be completed within a prescribed time regardless of initial conditions, motivating the adoption of prescribed-time stability (PTS) [18].To address this, prescribed-time control (PTC) [18] enables exact tracking-error convergence within a user-defined settling time through time-varying gains generated via state- or time-scaling techniques [19]. However, these gains become singular as time approaches the prescribed bound, and exogenous disturbances may induce uncertain post-convergence behavior. Smooth time-based generator (TBG) functions [20], [21], [22], [23] alleviate gain singularity, but their control synthesis generally requires prior knowledge of model parameters [10].

To address these challenges, we propose an adaptive barrier-function-based control policy, unlike approximation-based methods [24], [9], [25]. Existing approaches [7], [26], [27] typically enforce safety in Euler–Lagrange (EL) systems using prescribed performance functions. The proposed framework formulates a filtered error using TBG functions to prescribe settling time, synthesizes a state-constrained control law to ensure safety, and employs a saturation function to bound the control input, in contrast to the studies [28], [29], [30] that only impose state constraints. Moreover, the simultaneous involvement of state, input, and temporal (SIT) constraints reduces the domain of attraction [31], resulting in local PTS. Thus, the main contributions of this study are:

1) A unified approximation-free control framework that can be leveraged in both joint and task space tracking.
2) The proposed adaptive barrier function-based control policy achieves local prescribed time prescribed bound stability while adhering to SIT constraints.
3) The proposed adaptive control framework actively rejects any bounded exogenous while achieving superior tracking performance.

## II. METHODOLOGY

### A. Preliminaries

*Notations*: Bold letters denote matrices and vectors. The sets of real, non-negative real, and natural numbers are denoted by $\mathbb{R}$, $\mathbb{R}_{\geq 0}$, and $\mathbb{N}$, respectively. The vector inequalities, $\boldsymbol{a} \preceq \boldsymbol{b}$ $(\boldsymbol{a} \succeq \boldsymbol{b})$ imply $a_i \leq b_i$ $(a_i \geq b_i), i = \{1, \cdots, n\}$. $\boldsymbol{I}_n \in \mathbb{R}^{n \times n}$, and $\boldsymbol{0}_n, \boldsymbol{1}_n \in \mathbb{R}^n$ represent the identity matrix and vector of zeros and ones, respectively. $[\![\boldsymbol{a}, \boldsymbol{b}]\!] = [a_1, b_1] \times \cdots \times [a_n, b_n]$ denotes hyper interval, where $a_i$, $b_i$ are components of $\boldsymbol{a}$, $\boldsymbol{b}$, respectively. $\min(\boldsymbol{a}, \boldsymbol{b}) = [\min(a_i, b_i)]^\top$, $\max(\boldsymbol{a}, \boldsymbol{b}) = [\max(a_i, b_i)]^\top \ \forall i$. The Euclidean-norm is given by $\|\cdot\|$. $|\boldsymbol{a}| = [|a_1|, \cdots, |a_n|]^\top$.

Consider the class of EL systems with $\boldsymbol{q} : \mathbb{R}_{\geq 0} \to \mathbb{R}^n$ being position coordinates, $\dot{\boldsymbol{q}}$, $\ddot{\boldsymbol{q}}$ denote the first and second order derivatives of position coordinates, respectively, defined as below:

$$\boldsymbol{M}(\boldsymbol{q})\ddot{\boldsymbol{q}} + \boldsymbol{C}(\boldsymbol{q}, \dot{\boldsymbol{q}})\dot{\boldsymbol{q}} + \boldsymbol{g}(\boldsymbol{q}) + \boldsymbol{u}_d(t) = \boldsymbol{u}, \quad (1)$$

where $\boldsymbol{M}(\boldsymbol{q}) \in \mathbb{R}^{n \times n}$ is mass matrix, $\boldsymbol{C}(\boldsymbol{q}, \dot{\boldsymbol{q}}) \in \mathbb{R}^{n \times n}$ is Coriolis matrix, $\boldsymbol{g}(\boldsymbol{q}) \in \mathbb{R}^n$ is gravity vector, $\boldsymbol{u}_d(t) \in \mathbb{R}^n$ is external disturbance and $\boldsymbol{u} \in \mathbb{R}^n$ is the control input. For brevity, when a symbol's functional dependence is clear, its arguments and brackets are omitted; e.g., $\boldsymbol{C}(\boldsymbol{q}, \dot{\boldsymbol{q}})$ and $\boldsymbol{u}_d(t)$ are written as $\boldsymbol{C}$ and $\boldsymbol{u}_d$, respectively. Further, the properties of EL systems [32, Chapter 2], [33] are stated below for some positive real constants $\underline{M}, \overline{M}, \underline{m}, \overline{m}, \overline{C}, \overline{G}, \overline{F}$ that represent bounds on norm of the system matrices.

*Property 1:* The matrix $\dot{\boldsymbol{M}} - 2\boldsymbol{C}$ is skew-symmetric.
*Property 2:* $\underline{M}\boldsymbol{I}_n \leq \boldsymbol{M} \leq \overline{M}\boldsymbol{I}_n$ and $\underline{m}\boldsymbol{I}_n \leq \boldsymbol{M}^{-1} \leq \overline{m}\boldsymbol{I}_n$.
*Property 3:* $\|\boldsymbol{C}\| \leq \overline{C}\|\dot{\boldsymbol{q}}\|, \|\boldsymbol{G}\| \leq \overline{G}, \|\boldsymbol{F}\| \leq \overline{F}\|\dot{\boldsymbol{q}}\|$.

The state constraints on the system (1) are imposed as $\boldsymbol{q} \in [\![\underline{\boldsymbol{q}}, \overline{\boldsymbol{q}}]\!], \dot{\boldsymbol{q}} \in [\![\underline{\boldsymbol{\nu}}, \overline{\boldsymbol{\nu}}]\!], \forall t > 0$, where $\underline{\boldsymbol{q}}, \overline{\boldsymbol{q}}, \underline{\boldsymbol{\nu}}, \overline{\boldsymbol{\nu}}$ are known constants. Then, one can conclude that dynamic bounds[34] on velocity to enforce constraints on both position and velocity as $\underline{\boldsymbol{x}} = \max(\underline{\boldsymbol{\nu}}, \kappa(\underline{\boldsymbol{q}} - \boldsymbol{q})) \preceq \dot{\boldsymbol{q}} \preceq \min(\overline{\boldsymbol{\nu}}, \kappa(\overline{\boldsymbol{q}} - \boldsymbol{q})) = \overline{\boldsymbol{x}}$. Then, one can obtain $\boldsymbol{D}\dot{\boldsymbol{q}} \preceq [-\underline{\boldsymbol{x}}, \overline{\boldsymbol{x}}]^\top$, where $\boldsymbol{D} = [-\boldsymbol{I}_n, \boldsymbol{I}_n]^\top$. Then, introducing $\boldsymbol{y} : \mathbb{R}_{\geq 0} \to \mathbb{R}^{2n}$ for $i = 1, 2$, these inequality constraints become $\boldsymbol{D}\boldsymbol{q}(t) + \boldsymbol{y}(t) \odot \boldsymbol{y}(t) = [-\underline{\boldsymbol{x}}, \overline{\boldsymbol{x}}]^\top$, where $\odot$ denotes the Hadamard product. Then, reformulating this equality constraint using $\boldsymbol{\xi} : \mathbb{R}_{\geq 0} \to \mathbb{R}^6$ yields:

$$\boldsymbol{\xi}(t) = \boldsymbol{D}\boldsymbol{q}(t) + \boldsymbol{y}(t) \odot \boldsymbol{y}(t) - [-\underline{\boldsymbol{x}} - c, \overline{\boldsymbol{x}} - c]^\top, \quad (2)$$

where $c_i > 0$ is the safety margin. To evaluate $\boldsymbol{\xi}(t)$, we will later introduce an adaptive law on $\boldsymbol{y}(t)$. Let $\mathcal{X} \subset \mathbb{R}^n, \mathcal{U} \subset \mathbb{R}^m, \mathcal{W} \subset \mathbb{R}^p$ are compact subsets, a set $\mathcal{B}_\epsilon(\boldsymbol{x}_r) = \{\boldsymbol{x} \in \mathbb{R}^n : \|\boldsymbol{x} - \boldsymbol{x}_r\| \leq \epsilon\}$ denotes a ball of radius $\epsilon$ around the point $\boldsymbol{x}_r \in \mathbb{R}^n$, and $|\boldsymbol{a}|_\mathcal{C} = \inf_{\boldsymbol{b} \in \mathcal{C}} \|\boldsymbol{a} - \boldsymbol{b}\|$ for the set $\mathcal{C} \subset \mathbb{R}^n$. The following definition introduces local PTPB stability:

*Definition 1 ([35]):* For the system $\dot{\boldsymbol{x}} = \boldsymbol{f}(t, \boldsymbol{x}, \boldsymbol{u}, \boldsymbol{w})$ with $\boldsymbol{x} \in \mathcal{X}$, $\boldsymbol{u} = \mathbf{u}(t, \boldsymbol{x}) \in \mathcal{U}$, and $\boldsymbol{w} \in \mathcal{W}$, the equilibrium point $\boldsymbol{x}_r \in \mathcal{X}$ is said to be *local prescribed-time prescribed bound* (PTPB) stable for user-defined constants $T, \epsilon > 0$, if the closed-loop trajectory $\boldsymbol{\Psi}_\mathbf{u}(t, t_0, \boldsymbol{x}(t_0))$ remains in the ball $\mathcal{B}_\epsilon(\boldsymbol{x}_r)$ for any $\boldsymbol{x}(t_0) \in \mathcal{C} \subset \mathcal{X}$ after the prescribed time

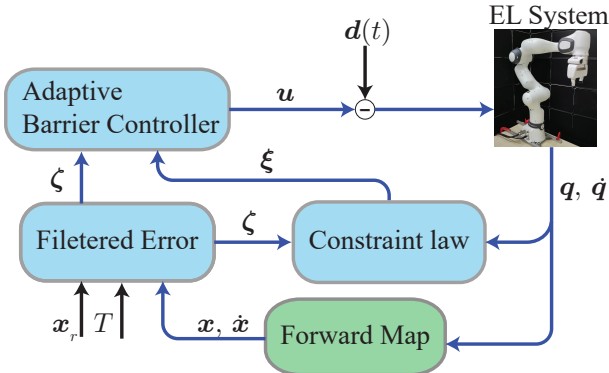

Fig. 1: Proposed Control framework

$T$, i.e., $|\boldsymbol{\Psi}_\mathbf{u}(t, t_0, \boldsymbol{x}(t_0))|_{\mathcal{B}_\epsilon(\boldsymbol{x}_r)} = 0 \ \forall \ t \geq t_0 + T$, where $\mathcal{C}$ is some neighbourhood of $\boldsymbol{x}_r$.

*Remark 1:* For system (1) subject to input saturation constraints, Definition 1 indicates that only trajectories initialized within a feasible set $\mathcal{C} \subset \mathbb{R}^n$ can be driven to the reference within the user-specified prescribed time. In particular, larger initial deviations demand greater control authority to ensure convergence within the prescribed interval [25]. A conservative approximation of the feasible set $\mathcal{C}$ can be obtained as demonstrated in [35].

Before stating our control objective, we first introduce the task-space formulation, inspired by the study [36], to achieve multiple tasks with equal priority for both redundant and non-redundant robotic manipulators. For a total $k \in \mathbb{N}$, let $\boldsymbol{x}_i : \mathbb{R}_{\geq 0} \to \mathbb{R}^{m_i}$ denote position and $\dot{\boldsymbol{x}}_i, \ddot{\boldsymbol{x}}_i$ denote corresponding time derivative, which are defined as follows:

$$\boldsymbol{x}_i = \boldsymbol{f}(\boldsymbol{q}), \ \dot{\boldsymbol{x}}_i = \boldsymbol{J}(\boldsymbol{q})\dot{\boldsymbol{q}}, \ \ddot{\boldsymbol{x}}_i = \boldsymbol{J}(\boldsymbol{q})\ddot{\boldsymbol{q}} + \dot{\boldsymbol{J}}(\boldsymbol{q})\dot{\boldsymbol{q}}, \quad (3)$$

for $i = 1, \cdots, k$ with each task dimension is $m_i$ such that $\sum m_i = n$ and $\boldsymbol{f}_i : \mathbb{R}^n \to \mathbb{R}^{m_i}$ are continuously differentiable function, $\boldsymbol{J}_i(\boldsymbol{q}) = \partial \boldsymbol{f}_i(\boldsymbol{q})/\partial \boldsymbol{q}$. Then, one can augment the forward map and its Jacobian as follows:

$$\boldsymbol{f} = [\boldsymbol{f}_1^\top, \cdots, \boldsymbol{f}_k^\top]^\top, \boldsymbol{x} = [\boldsymbol{x}_1^\top, \cdots, \boldsymbol{x}_k^\top]^\top \quad (4)$$

$$\boldsymbol{J} = [\boldsymbol{J}_1^\top, \cdots, \boldsymbol{J}_k^\top]^\top, \dot{\boldsymbol{J}} = [\dot{\boldsymbol{J}}_1^\top, \cdots, \dot{\boldsymbol{J}}_k^\top]^\top. \quad (5)$$

Then, using the above formulation, one can reformulate the system dynamics into task space dynamics as follows:

$$\boldsymbol{M}_x\ddot{\boldsymbol{x}} + \boldsymbol{C}_x\dot{\boldsymbol{x}} + \boldsymbol{g}_x + \boldsymbol{u}_{dx} = (\boldsymbol{J})^{-1}\boldsymbol{u}, \quad (6)$$

where $\boldsymbol{M}_x = (\boldsymbol{J}\boldsymbol{M}^{-1}\boldsymbol{J}^\top)^{-1}, \boldsymbol{C}_x = (\boldsymbol{J}^\top)^{-1}\boldsymbol{C}\boldsymbol{J}^{-1} - \boldsymbol{M}_x\dot{\boldsymbol{J}}\boldsymbol{J}^{-1}, \boldsymbol{g}_x = \boldsymbol{J}^{-1}\boldsymbol{g}, \boldsymbol{u}_{dx} = \boldsymbol{J}^{-1}\boldsymbol{u}_d$. Let $\boldsymbol{x}_r(t), \dot{\boldsymbol{x}}_r(t)$, and $\ddot{\boldsymbol{x}}_r(t)$ denote reference position, velocity, and acceleration trajectory, respectively. The following assumptions are made in this study.

*Assumption 1:* External disturbances are bounded, i.e. $\|\boldsymbol{u}_d\| \leq \overline{u}_d$.

*Assumption 2:* The reference trajectories $\boldsymbol{x}_r(t), \dot{\boldsymbol{x}}_r(t)$ satisfy state constraints, and $\ddot{\boldsymbol{q}}_r(t)$ is bounded.

*Assumption 3:* The forward map is known, and the Jacobian $\boldsymbol{J}$ is of full row rank. For some unknown constants,

$\|\boldsymbol{J}(\boldsymbol{q})\| \leq \overline{J}_1, \|\boldsymbol{J^{-1}}\| \leq \overline{J}_2, \|\partial \boldsymbol{J}/\partial q_i\| \leq \overline{J}_{3,i}, \forall i = \{1, \cdots, n\}, \overline{J}_3 = \max_i \overline{J}_{3,i}$.

*Problem:* For the given robotic manipulator system in (1) subjected to *Properties* 1–3 and *Assumptions* $1-3$, the control objective is to synthesize a control policy that achieves local PTPB stability as in *Definition* 1 while adhering to SIT constraints in the presence of exogenous disturbances.

*B. Adaptive Barrier Control Policy*

Firstly, define error vectors as $\boldsymbol{e}(t) = \boldsymbol{x}(t) - \boldsymbol{x}_r(t)$. Then, using TBG functions $h_1(t), h_2(t)^2$, define $\boldsymbol{\varepsilon}(t) = \boldsymbol{e}(t) - \boldsymbol{r}(t)$ with $\boldsymbol{r}(t) = h_1(t)\boldsymbol{e}(0) + h_2(t)\dot{\boldsymbol{e}}(0)$, for ensuring convergence in time $T_i$.

$$\boldsymbol{\zeta}(t) = \dot{\boldsymbol{\varepsilon}}(t) + \boldsymbol{K}\boldsymbol{\varepsilon}(t), \tag{7}$$

$\boldsymbol{K} \in \mathbb{R}^{n \times n}$ is diagonal with positive entries. Then, input constraints on $\boldsymbol{u}$ are enforced via saturation function: $\boldsymbol{u} = \boldsymbol{S}(\boldsymbol{\tau})\boldsymbol{\tau}$, where $\boldsymbol{S}(\boldsymbol{\tau}) = \mathrm{diag}\,(S_1(\tau_1), \cdots, S_n(\tau_n))$.

$$S_i(\tau_i(t)) = \mathrm{med}\left(\frac{\underline{u}_i}{\tau_i}, 1, \frac{\overline{u}_i}{\tau_i}\right),$$

where $\mathrm{med}(a, b, c)$ is the median function, $\tau_i$ is the $i^{\mathrm{th}}$ component of $\boldsymbol{\tau}$, $\overline{\boldsymbol{u}} \succeq 0$ and $\underline{\boldsymbol{u}} \preceq 0$ are known constants such that $\boldsymbol{u}_m \in [\![\underline{\boldsymbol{u}}, \overline{\boldsymbol{u}}]\!]$, with $\underline{u}_i$ and $\overline{u}_i$ denoting the components of $\underline{\boldsymbol{u}}$ and $\overline{\boldsymbol{u}}$. Let $\rho, \omega, \alpha, \beta > 0$ be user-defined constants. Then, control policy $\boldsymbol{\tau}$, is designed as follows:

$$\dot{\boldsymbol{y}} = (2\mathrm{diag}(\boldsymbol{y}))^{-1}\left(-\beta\boldsymbol{\xi} + \beta\alpha\boldsymbol{D}\boldsymbol{\zeta} - \boldsymbol{D}\dot{\boldsymbol{\zeta}}\right), \tag{8}$$

$$\boldsymbol{\tau} = -a(t)(\gamma(\boldsymbol{\varepsilon}, \dot{\boldsymbol{\varepsilon}}) + \|\boldsymbol{D}\|\|\boldsymbol{\xi}\|)\boldsymbol{W}\lceil\boldsymbol{\zeta}(t)\rceil^0,$$

$$a(t) = \frac{\rho\|\boldsymbol{\zeta}\|}{\omega - \|\boldsymbol{\zeta}\|}, \quad \lceil\boldsymbol{\zeta}\rceil^0 = \frac{\boldsymbol{\zeta}}{\|\boldsymbol{\zeta}\|}, \tag{9}$$

where $\gamma(\boldsymbol{\varepsilon}, \dot{\boldsymbol{\varepsilon}}) = 4\max(1, \|\boldsymbol{\varepsilon}\|, \|\dot{\boldsymbol{\varepsilon}}\|, \|\boldsymbol{\varepsilon}\|\|\dot{\boldsymbol{\varepsilon}}\|)$, and $\boldsymbol{\xi}(0) = \boldsymbol{0}_{2n}$. The Fig. 1 illustrates the proposed control framework in (9).

*Theorem 1:* Consider the robotic manipulator system (1) with *Properties* 1–3, *Assumptions* $1-3$, and sufficiently large $T$. The synthesized controller policy (9) achieves robust local PTPB stability as in *Definition* 1 while adhering to SIT constraints subject to exogenous disturbances with bound $\epsilon = \sqrt{(\omega/\underline{K})^2 + \omega^2(1 + \overline{K}/\underline{K})^2}$.

For proof, see Appendix I. Note that $\overline{K}_i$ ($\underline{K}_i$) denote the maximum (minimum) diagonal entry of $\boldsymbol{K}_i$.

*Remark 2:* The proposed control policy is applicable to both joint- and task-space schemes. Notably, when all $k$ tasks are defined in joint space, $\boldsymbol{f}_i(\boldsymbol{q})$ and $\boldsymbol{J}_i(\boldsymbol{q})$ reduce to the identity maps and matrices, respectively. Moreover, the strategy is model-parameter-free in (1), but it assumes the forward mapping $\boldsymbol{f}_i(\boldsymbol{q})$ is known for task-space measurements.

*Remark 3:* Notice that the proposed control framework (9) rejects external disturbances without disturbance measurements, adaptive laws, or observers, unlike existing approaches [8], [16], [3], [4]. This is achieved by leveraging the boundedness of the lumped uncertainty under the imposed system *Properties* and *Assumptions*, while the adaptive barrier gain $a(t)$ increases as needed to compensate for

$^2$are smooth piecewise fifth-order polynomials; see [35]

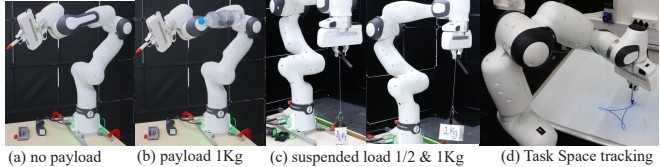

(a) no payload    (b) payload 1Kg    (c) suspended load 1/2 & 1Kg    (d) Task Space tracking

Fig. 2: Illustration of experimental setup for Franka Research 3 robotic arm for various payload conditions.

disturbances without becoming unbounded (see *Theorem* I). Consequently, the adaptive gains remain bounded and non-singular, ensuring robustness and adaptability in uncertain environments.

## III. Results and Discussion

This section details experimental results[3] on a 7-DoF Franka Research 3 (FR3) robotic manipulator under various conditions. All computations are carried out on a Dell Laptop with 16 GB of RAM and Ubuntu 22.04 running on a real-time Linux kernel.

The choice of control policy parameters is omitted due to page limitations. In this experimental study, we have considered the following scenarios (see Fig. 2):

a) a quintic trajectory tracking in joint space with/without an external payload with $T = 4s$ (see Fig. 3)
b) task space tracking with the following subtasks and $T = 2s$ ((see Fig. 4)).
   i) hypotrochoid curve tracking in Cartesian space
   ii) posture alignment with always z-axis up using the last row of the rotation matrix, i.e., $\boldsymbol{x}_2 = [r_{3,1}, r_{3,2}]^\top$
   iii) a sinusoidal curve for joint 3 and joint 7 $q_{i,0} + 0.01\sin(2\pi t/10)$, where $i = 3, 7$
c) tracking in task space while carrying suspended loads of 0.5Kg ad 1Kg with similar subtasks as above, except for the tracking point-to-point in cartesian space. See Fig. 5.

## IV. Conclusion

In this brief, we proposed an adaptive barrier-function-based control policy for robotic manipulators, applicable to both joint- and task-space tracking while rejecting exogenous disturbances. A TBG-based filtered error, together with state constraints and input saturation, ensures safe control under bounded inputs. Stability is established through Lyapunov analysis and validated experimentally on a 7-DoF robotic arm. Future work will extend the framework to unified motion and force control in uncertain environments.

## Appendix I
### Proofs for *Theorem*

Let $\underline{M}_x, \overline{M}_x, \underline{m}_x, \overline{m}_x, \overline{C}_x, \overline{u}_{dx} > 0$ be some known constants. Following the *Properties* 1–3 and *Assumption* 1–3, one can obtain bounds on system matrices in (6) $\|\boldsymbol{M}_x\| \leq \overline{M}_x$, $\|\boldsymbol{C}_x\| \leq \overline{C}_x$, $\|\boldsymbol{g}_x\| \leq \overline{g}_x$, and $\|\boldsymbol{u}_{dx}\| \leq \overline{u}_{dx}$, where $\overline{M}_x = \overline{J}_2^2 \overline{M}, \overline{C}_x = \overline{J}_2^2(\overline{C} + \overline{M}\,\overline{J}_2\overline{J}_3), \overline{g}_x =$

$^3$Video link: https://drive.google.com/file/d/1w_kO3HgBdsdw1wD-Z0hvKNy_2b8zpg2q/view?usp=drive_link

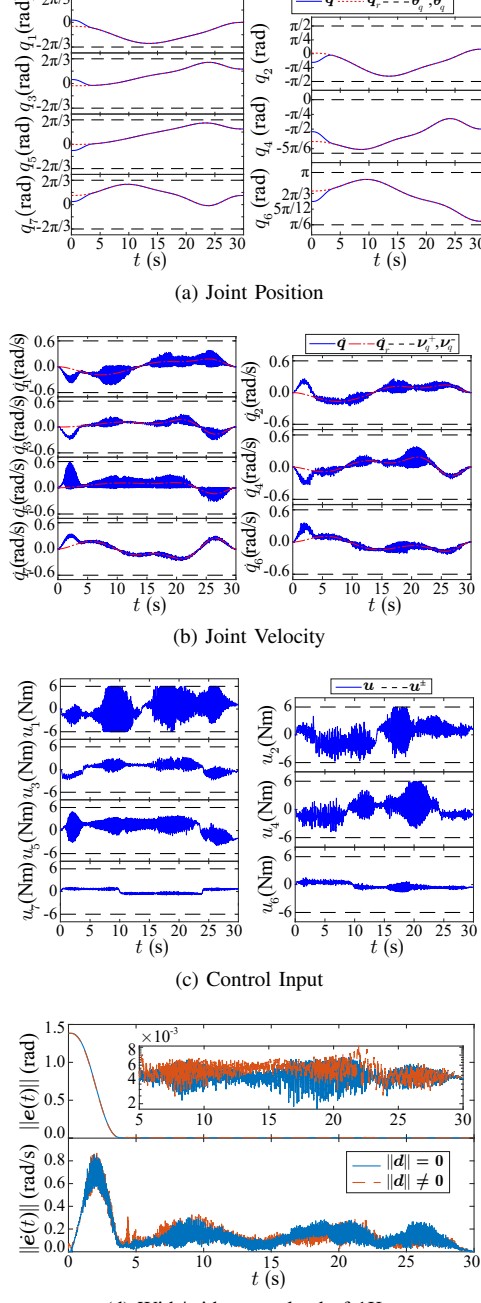

(a) Joint Position

(b) Joint Velocity

(c) Control Input

(d) With/without payload of 1Kg

Fig. 3: Experimental results for the FR3 tracking a reference trajectory in joint space (Case a) with an initial offset of $30°$ in each joint, demonstrating state and input constraint satisfaction under the proposed control policy (9).

$\overline{J}_2\overline{g}$, and $\overline{u}_{dx} = \overline{J}_2\overline{u}_d$, respectively. Further, the error dynamics can be written as $\dot{\zeta}$ with $W = J^{-1}$ as in (10).

$$M_x\dot{\zeta} = W^\top u - C_x\zeta + \delta_x, \qquad (10)$$

where $\delta_x(t) = C_x(\zeta - \dot{x}) - g_x - u_{dx} - M_x(-\ddot{x}_r - \ddot{e}_1 + K\dot{\varepsilon})$. Then, the boundedness of uncertainty is derived as follows:

$$\|\delta_x(t)\| \leq \overline{\delta}\gamma(\varepsilon, \dot{\varepsilon}), \qquad (11)$$

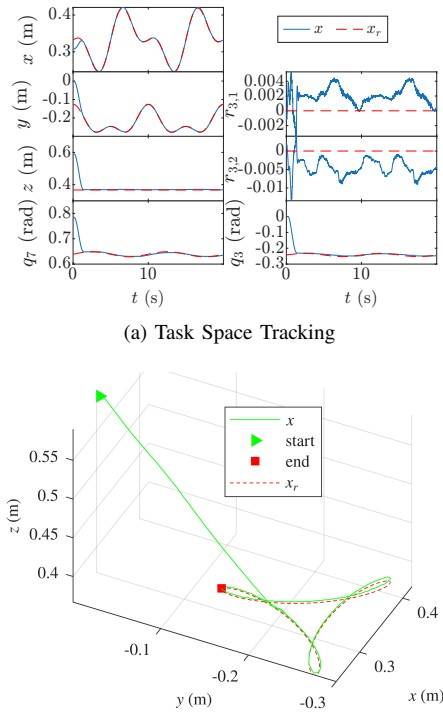

(a) Task Space Tracking

(b) Cartesian space (first three coordinates)

Fig. 4: Experimental results for the FR3 robotic arm tracking a reference trajectory in task space (Case b) with $T = 1.5$s and an offset of 0.26m in for task 1 (b.i)

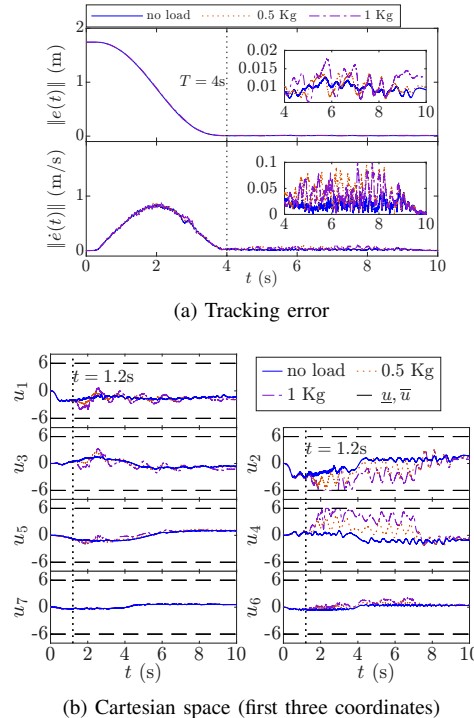

(a) Tracking error

(b) Cartesian space (first three coordinates)

Fig. 5: Experimental results for the FR3 tracking a task-space point-to-point trajectory (Case c) with a suspended load of $0.5, 1$ Kg, $T = 4$s, and a 1.7m position offset. Around $t = 1.2$s, the suspended load tension switches from slack to tight.

where $\overline{\delta} = \max\left(\overline{\delta}_1, \overline{\delta}_2, \overline{\delta}_3, \overline{\delta}_4\right)$, $\overline{\delta}_1 = \overline{M}_x(\overline{x}_{2d} + \overline{e}_{2d}) + \overline{C}_x(\overline{e}_{1d} + \overline{x}_{1d})^2 + \overline{g}_x + \overline{u}_{dx}$, $\overline{\delta}_2 = \overline{C}_x\|\boldsymbol{K}\|(\overline{e}_{1d} + \overline{x}_{1d})$, $\overline{\delta}_3 = \overline{C}_x(\overline{e}_{1d} + \overline{x}_{1d}) + \overline{M}_x\|\boldsymbol{K}\|$ and $\overline{\delta}_4 = \overline{C}_x\|\boldsymbol{K}\|$

*Proof:* Consider the Lyapunov function candidate $V(t) = \frac{1}{2}\boldsymbol{\zeta}^\top \boldsymbol{M}_x \boldsymbol{\zeta} + \frac{1}{2\beta}\boldsymbol{\xi}^\top \boldsymbol{\xi}$ and from (8) one can derive $\dot{\boldsymbol{\xi}} = -\beta\boldsymbol{\xi} + \beta\alpha\boldsymbol{D}\boldsymbol{\zeta}$. Then, taking the time derivative of $V(t)$ and using the EL properties, we get:

$$\dot{V} = -\frac{\rho\gamma}{\omega - \|\boldsymbol{\zeta}\|}\boldsymbol{\zeta}^\top \boldsymbol{W}^\top \boldsymbol{S}\boldsymbol{W}\boldsymbol{\zeta} + \boldsymbol{\zeta}^\top \boldsymbol{\delta}$$
$$-\frac{\rho\|\boldsymbol{D}\|\|\boldsymbol{\xi}\|}{\omega - \|\boldsymbol{\zeta}\|}\boldsymbol{\zeta}^\top \boldsymbol{W}^\top \boldsymbol{S}\boldsymbol{W}\boldsymbol{\zeta} - \boldsymbol{\xi}^\top \boldsymbol{\xi} + \alpha\boldsymbol{\xi}^\top \boldsymbol{D}\boldsymbol{\zeta}. \quad (12)$$

Now, from *Assumption* 1, there exists $r>0$ such that $\boldsymbol{\zeta}^\top \boldsymbol{W}^\top \boldsymbol{S}\boldsymbol{W}\boldsymbol{\zeta} \geq r\|\boldsymbol{\zeta}\|^2$, where $r = \lambda_{\min}(\boldsymbol{W}^\top \boldsymbol{S}\boldsymbol{W})$. Since, $\gamma > 4$ and from (11) and (12), we have

$$\dot{V} \leq 4\left(\frac{r\rho\|\boldsymbol{\zeta}\|}{\omega - \|\boldsymbol{\zeta}\|} - \overline{\delta}\right)\|\boldsymbol{\zeta}\|$$
$$-\left(\frac{r\rho\|\boldsymbol{\zeta}\|}{\omega - \|\boldsymbol{\zeta}\|} - \alpha\right)\|\boldsymbol{D}\|\|\boldsymbol{\xi}\|\|\boldsymbol{\zeta}\|. \quad (13)$$

Then, provided $\|\boldsymbol{\zeta}\| > \max\left(\frac{\omega\overline{\delta}}{r\rho + \overline{\delta}}, \frac{\omega\alpha}{r\rho + \alpha}\right)$, results in $\dot{V}(t) \leq 0$. Since $\boldsymbol{\zeta}(0) = \boldsymbol{0}_n$ from (7), we have

$$\|\boldsymbol{\zeta}\| < \max\left(\frac{\omega\overline{\delta}}{r\rho + \overline{\delta}}, \frac{\omega\alpha}{r\rho + \alpha}\right) < \omega. \quad (14)$$

Also, from (7), we have $\dot{\boldsymbol{e}} + \boldsymbol{K}\boldsymbol{e} = \boldsymbol{\zeta} + \dot{\boldsymbol{r}} + \boldsymbol{K}\boldsymbol{r}$. Since, $\|\boldsymbol{\zeta}\| < \omega, \|\boldsymbol{r}(t)\| = 0, \forall t \geq T$, solving the ODE yields $\|\boldsymbol{e}\| < \omega/\underline{K}$ and $\|\dot{\boldsymbol{e}}\| \leq \omega(1 + \overline{K}/\underline{K})$, resulting in the prescribed bound $\epsilon$. Further, from $\|\boldsymbol{\zeta}\| < \omega$ and (8) with $\alpha < c/(\|D\|\omega)$ imply $\|\boldsymbol{\xi}\| < c$ using. Thus, from (2) we have $\boldsymbol{q} \in [\![\underline{\boldsymbol{q}}, \overline{\boldsymbol{q}}]\!], \dot{\boldsymbol{q}} \in [\![\underline{\boldsymbol{\nu}}, \overline{\boldsymbol{\nu}}]\!]$. Hence, the control policy (9) achieves robust local PTPB stability under SIT constraints. $\blacksquare$

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
