# OpenReview forum: "Robust Control of Robotic Manipulator with Prescribed Spatio-temporal and Input constraints"
_IEEE.org/ICRA/2026/Workshop/Manipulation_Robustness — ICRA 2026_

### Official Review · Reviewer_f7CR · 2026-05-09
**A promising and timely framework for robust constrained robotic manipulation, with clearer validation needed to strengthen its impact.**

**Rating:** 7
**Confidence:** 3

**Review:**

The manuscript proposes a robust adaptive control framework for robotic manipulators that achieves prescribed-time convergence while enforcing simultaneous state, input, and temporal (SIT) constraints under bounded disturbances. The topic is relevant to manipulation robustness under real-world constraints, and the combination of time-based generator functions, adaptive barrier control, and input saturation is conceptually interesting. However, the presentation is dense, with heavy notation and unclear structural separation between system dynamics, constraints, and control design, which makes the overall methodology difficult to follow.

The theoretical development relies on several boundedness assumptions and conservative inequalities. While standard in the literature, their practical restrictiveness and implications for real-world robotic deployment are not sufficiently discussed. In particular, the role of auxiliary variables and gain-dependent conditions is not clearly connected to controller tuning, and the impact of approximation steps on conservatism is missing. Given the workshop focus on manipulation robustness under real-world challenges, a clearer discussion of how these constraints and assumptions affect practical applicability is essential.

Experimentally, results on a 7-DoF Franka arm under disturbances are promising but remain limited in scope. The absence of comparisons with representative baselines makes it difficult to assess improvements in robustness relative to existing methods, which is central to the workshop theme. Minor issues include unclear notation, repeated video links, and formatting inconsistencies (reference 32 has a "chapter 2"). Overall, while the paper addresses an interesting constrained control problem, its connection to real-world manipulation robustness is not yet convincingly demonstrated, and both theoretical clarity and experimental validation need strengthening.

In summary, several smaller issues should also be addressed:
- Some claims are broad or insufficiently justified, such as the assertion that the method achieves “superior tracking performance” without quantitative comparison.
- The manuscript would benefit from clearer notation and more consistent definitions, especially in the transition between joint-space and task-space formulations.
- There are minor typographical and formatting issues. The video link is shown several times in the paper. The reference 32 has a chapter 2 along with it.
- The real-world applications need more clarification about how it satisfies the limitations of the constrained parameters and how it actually performs when compared to other methods.

Overall, the paper presents an interesting integration of prescribed-time control with constraint-handling strategies for robotic manipulators. However, the manuscript would benefit from clearer presentation, more precise and consistent problem definition, and a more comprehensive experimental evaluation to better substantiate the proposed method’s effectiveness.

---

### Decision · Program_Chairs · 2026-05-21

Accept